# Challenges and Future of Drug-Induced Liver Injury Research—Laboratory Tests

**DOI:** 10.3390/ijms23116049

**Published:** 2022-05-27

**Authors:** Sabine Weber, Alexander L. Gerbes

**Affiliations:** Department of Medicine II, Liver Centre Munich, LMU Klinikum Munich, 81377 Munich, Germany; gerbes@med.uni-muenchen.de

**Keywords:** drug-induced liver injury, hepatotoxicity, biomarkers, drug development, adverse drug events

## Abstract

Drug-induced liver injury (DILI) is a rare but potentially severe adverse drug event, which is also a major cause of study cessation and market withdrawal during drug development. Since no acknowledged diagnostic tests are available, DILI diagnosis poses a major challenge both in clinical practice as well as in pharmacovigilance. Differentiation from other liver diseases and the identification of the causative agent in the case of polymedication are the main issues that clinicians and drug developers face in this regard. Thus, efforts have been made to establish diagnostic testing methods and biomarkers in order to safely diagnose DILI and ensure a distinguishment from alternative liver pathologies. This review provides an overview of the diagnostic methods used in differential diagnosis, especially with regards to autoimmune hepatitis (AIH) and drug-induced autoimmune hepatitis (DI-AIH), in vitro causality methods using individual blood samples, biomarkers for diagnosis and severity prediction, as well as experimental predictive models utilized in pre-clinical settings during drug development regimes.

## 1. Introduction

Drug-induced liver injury (DILI) is an infrequent yet potentially life-threatening adverse drug reaction, which accounts for the majority of acute liver failure cases in Western societies [1,2]. DILI is also one of the leading causes of drug attrition during clinical studies and of market withdrawal after initial approval [3]. Thus, despite its rare occurrence, DILI represents a major health and economical problem highlighting that the rapid detection, prognosis of severity and possibly also prediction of DILI cases are of utmost importance.

DILI should be considered when a relevant liver enzyme elevation occurs after drug exposure, as defined by the 2011 consensus criteria: alanine aminotransferase (ALT) ≥ 5 times the upper limit of normal (ULN), alkaline phosphatase (ALP) ≥ 2 × ULN or ALT ≥ 3 × ULN and total bilirubin (TBIL) > 2 × ULN [4]. Liver enzyme elevation and temporal relationship with drug intake remain the hallmark in DILI diagnosis since, so far, no single method has been established to diagnose DILI safely and accurately. However, the accuracy of liver enzymes for DILI diagnosis is poor. ALT and aspartate aminotransferase (AST) elevation can also be induced by other conditions apart from liver damage, such as muscle injury [5,6], while ALP elevation also occurs due to liver-unrelated pathologies, such as bone pathologies, renal dysfunction, acute inflammatory diseases or malignancies [7]. Moreover, none of the liver parameters mentioned above are specific to a certain type of liver injury. Apart from limitations due to specificity, poor sensitivity of aminotransferase elevation in the diagnosis of DILI is an additional limiting factor. Transient ALT elevations without clinical significance, spontaneously disappearing despite drug continuation, can occur, a phenomenon called adaptation which can also lead to attrition or delays in drug development [8]. Thus, the currently used laboratory tests in clinical practice lack diagnostic accuracy, cannot adequately distinguish between different types of liver injury, do not associate well with histopathological severity gradings and have low prognostic value [9]. Therefore, major efforts have been put into the development of diagnostic tests and the identification of biomarkers or biomarker panels [10]. In addition, prediction of DILI, in particular meaning the identification of novel drugs with high DILI potential early during drug development, has gained higher attention in recent years.

## 2. Differential Diagnosis—Causality Assessment Methods

The clinical picture of DILI is highly variable and can mimic different liver diseases [11]. DILI is a diagnosis of exclusion, other possible reasons for liver injury need to be evaluated and excluded. DILI diagnosis therefore remains a clinical challenge. Due to the lack of diagnostic tests, the current diagnostic standard is based on expert consensus opinion [12]. In addition, the Roussel Uclaf Causality Assessment Method (RUCAM) scale can serve as a supporting causality assessment tool (Table 1) [13]. However, the RUCAM scale has major limitations due to high subjectivity and inter-observer variability [14,15]. The application of the RUCAM is especially limited with regards to polymedicated patients, novel therapeutic agents, for which hepatotoxic potential and signature of typical liver injury patterns are largely unknown, herbal and dietary supplements and oncological therapeutics with alternative application schemes, e.g., immune checkpoint inhibitors. In particular in polymedicated patients, applying the RUCAM can be problematic since it should be calculated for all medications in question. The simultaneous treatment with other drugs will lead to a lower score of the others and vice versa. Moreover, drugs that are more commonly associated with DILI will reach higher total scores, since more points will be given for known hepatotoxicity. The limitations of the RUCAM scale and also other diagnostic algorithms, such as the Digestive Disease Week Japan (DDW-J) scale or the Maria and Victorino scale, might be overcome by combining those scores with pathological patterns [16]. Nevertheless, also regarding histomorphological characteristics, great overlaps with other liver diseases have been described [17,18], impeding the use of pathological patterns to safely diagnose or exclude DILI. To overcome these limitations, a revised electronic version of the RUCAM, the so-called RECAM (revised electronic causality assessment method), has recently been proposed (Appendix A) [19]. In the RECAM, risk factors and concomitant drugs were removed from the scoring system; moreover, the previous information on drug hepatotoxicity was linked to the LiverTox® likelihood scales [20] promoting higher objectivity. In addition, other causes of liver injury, which had not been regarded in the RUCAM, such as hepatitis E or infiltrating cancer, were added as potential non-drug causes. While the areas under the receiver operator curve (AUROCs) for the identification of probable DILI or higher were the same for the RUCAM and RECAM and the correlations between expert opinion and the diagnostic categories resulting from the RUCAM and RECAM were highly similar for both scores, the RECAM showed a better performance regarding the detection of extreme DILI diagnostic categories, i.e., highly likely/probable or unlikely/excluded [19].

## 3. Drug-Induced Autoimmune Hepatitis and Autoimmune Hepatitis Differences and Overlaps in Presentation and Diagnosis

Differentiating DILI from autoimmune hepatitis (AIH) is a particular challenge since major similarities between both entities are rather frequent. Both DILI and AIH patients can present with autoantibodies and/or elevation of immunoglobulins [21,22,23]. Moreover, DILI with autoimmune features and association with certain drug classes has also been proposed as a different disease entity, often referred to as drug-induced AIH (DI-AIH) [21,22,24]. DI-AIH has been assumed when elevated immunoglobulin G levels and/or titers of antinuclear antibodies (ANA) are detected in such DILI cases [21,23]. DI-AIH is thought to develop due to formation of neoantigens through drug–protein adducts with consecutive induction of immune activation. Furthermore, drugs can induce an immune imbalance between effector and regulatory lymphocytes and therefore can lead to a loss of self-tolerance with induction of autoimmune liver injury, which may persist beyond the cessation of the drug [24]. However, thus far, no clear definition for DI-AIH has been established. Importantly, it has also been demonstrated that ANA, a hallmark in the diagnosis of AIH [25,26] and the autoantibodies most often associated with DI-AIH [27,28], frequently occur in DILI patients without being associated with a specific clinical phenotype nor a certain outcome [29]. The role of immunoglobulin G (IgG) in DILI has not been evaluated systematically; however, high rates of IgG elevation in DILI patients have been reported repeatedly [3,21,29]. In addition, anti-mitochondrial antibodies (AMA), usually associated with primary biliary cholangitis, have been described in DILI patients and patients with acute liver failure due to variate causes including DILI [29,30,31]. Interestingly, the presence of AMA has been shown to be associated with a more severe form of DILI [29]. Thus, it can be speculated that rather than being specific for a certain type of liver injury or a specific DILI phenotype, autoantibodies might indicate severity of liver injury with more severe cases showing higher positivity rates for AMA in particular, while the occurrence of ANA seems to only represent an epiphenomenon of the acute DILI episode.

## 4. Differential Diagnosis—Use of In Vitro Assessment Tools

In order to improve DILI diagnosis and differentiation from other liver diseases, the use of the lymphocyte transformation test (LTT) has been proposed. Taking into account the role of the adaptive immune system, in particular T-cell activation, during the development of DILI, the LTT analyses whether a lymphocyte sensitization against a specific agent has occurred [32]. The LTT is an ex vivo test, for which peripheral blood mononuclear cells (PBMCs) are incubated with the respective drugs followed by the measurement of lymphocyte proliferation [32]. The LTT is based on the measurement of 3H-thymidine or interferon-gamma (IFN-γ) production as markers for T-cell proliferation [33]. However, the LTT has not been shown to adequately diagnose a specific drug reaction [34]. A possible approach to enhance the sensitivity of the LTT is to add prostaglandin inhibitors to the stimulation assay since prostaglandin-producing suppressor cells are thought to dampen the detection of T-cell activation upon drug stimulation in standard conditions [33,35]. Moreover, a modified version of the LTT has been proposed in 2017 measuring the production of granzyme B and specific cytokines, such as interleukin-2 (IL-2), IL-5, IL-13 and IFN-γ, after incubation of PBMCs with the possibly causative drugs. Unfortunately, this modified LTT failed to demonstrate a lymphocyte sensitization against the implicated drugs, and positive tests were only observed in two cases of isoniazid DILI [36]. The high rates of false positive results might be due to DILI being caused by reactive metabolites rather than by the parent drugs themselves [37]. Because of the low sensitivity and specificity, the LTT is seldomly used and has in fact not been standardized nor validated. Yet, despite these major limitations, the LTT is an integral component of the DDW-J 2004 scale, an assessment tool similar to the RUCAM only utilized in Japan [38]. However, while the DDJ-W 2004 has been described and compared to the RUCAM in several English publications, it has not been validated in English as yet [16,38,39].

In addition to the LTT, cluster of differentiation 69 (CD69) upregulation as a marker for T-cell activation has been proposed as a peripheral blood marker for drug hypersensitivity. It was shown in 15 patients that all patients with positive LTT also showed an increase in CD69 expression on T cells after incubation with the implicated drugs, while five healthy controls did not exhibit CD69 upregulation [40]. However, drug hypersensitivity was not limited to the liver, and validation studies have not been published.

Another in vitro causality assay has been proposed using monocyte-derived hepatocyte-like (MH) cells [41]. By incubating MH cells with possible causative agents, potential individual susceptibility toward drug hepatotoxicity has been assessed. Briefly, monocytes are isolated from the patients’ blood and cultivated under serum-free conditions for 10 days leading to the generation of cells with several hepatocyte features, in particular cytochrome P450 (CYP) activity. These MH cells are then incubated with the implicated drugs at a concentration of 1 × C_max_ and 10 × C_max_ for 48 h, and toxicity is analyzed using a standardized algorithm measuring the release of lactate dehydrogenase in the cell lysates and supernatant (Figure 1) [42]. The internal validation of this MH cell test was performed in a cohort of patients with positive and negative drug re-exposure, which is one of the most secure indications for drug hepatotoxicity in a given subject [3]. Toxicity was proven in 12 of 13 patients (92%) with positive re-exposure, while the test yielded negative results in all of the 27 patients with negative re-exposure [43]. However, the MH cell test has not been validated externally thus far.

## 5. Biomarkers in DILI—Promising Diagnostic Candidates

The difficulties in correctly diagnosing DILI and identifying the causative medication in polymedicated patients underline the importance of the detection of biomarkers or biomarker panels for the diagnosis of DILI in clinical practice. In the search for diagnostic biomarkers for drug hepatotoxicity, most efforts have initially been put into the identification of mechanistic biomarkers. Since reactive drug metabolites, mitochondrial damage and necrosis are thought to play a major role in DILI development [44,45], markers such as damage-associated molecular pattern molecules (DAMPS), heat shock proteins, adenosine triphosphate (ATP), high-mobility group box 1 (HMGB1) or markers of mitochondrial damage, e.g., glutamate dehydrogenase (GLDH), nuclear deoxyribonucleic acid (DNA) fragments or micro-RNA (miR), were initially sought after (Table 2) [46].

One particular **miR**, miR-122, has been repeatedly associated with DILI [47,48,49,50]. Being a liver-enriched miR, elevation of miR-122 has been observed even prior to ALT elevation in patients with paracetamol overdose [51]. In addition, the combination of miR-122, HMGB1 and keratin 18 (K18) has been shown to perform better in predicting paracetamol-induced liver injury than ALT alone [52]. However, this has been challenged by a recent study demonstrating that in DILI patients, GLDH correlated better with ALT than miR-122 and that miR-122 showed a pronounced inter- and intra-individual variability [53]. miR-192, another miR enriched in the liver, has also been proposed as an early predictor for DILI [54]. However, miR-192 has additionally been described as a marker for progression of non-alcoholic fatty liver disease (NAFLD), questioning the specificity of miR-192 for DILI diagnosis [55]. While single-miRNA expression levels can be highly similar in patients with DILI and alternative liver diseases, it was shown that the separation from liver injury and control subjects, as well as a differentiation of different liver injury types, was possible when a profile of 37 miRNA was applied [56]. The application of miRNA profiles as biomarker panels should therefore be evaluated in larger prospective studies.

The **K18** epitope M65 as a marker for necrosis and apoptosis was also found elevated earlier and to a higher degree than ALT in paracetamol-induced liver injury. In addition, M65 levels already declined after treatment withdrawal when ALT still remained high, indicating that K18, and in particular its epitope M65, could be a good follow-up marker in paracetamol DILI [50].

As mentioned above, GLDH has been shown to correlate well with ALT in DILI [53] and might therefore be a promising liver-specific biomarker. The specificity for the liver was further demonstrated in a study comparing ALT and GLDH levels in patients with Duchene muscular dystrophy and healthy controls. While ALT levels were increased 20-fold in the patients with Duchene muscular dystrophy, GLDH levels were comparable between patients and healthy controls [57]. GLDH has certain strengths that increase its potential as a biomarker since it is influenced only to a low degree by age and gender and demonstrates little intra- and inter-subject variability [58]. However, GLDH elevations might also occur due to common bile duct stone passage, circulatory problems resulting in ischemic or congestive hepatitis and due to some drugs not associated with clinically relevant DILI, such as colestyramine or heparins [58,59].

Recently, it was also demonstrated that a combined model comprising GLDH, K-18 and miR-122 was able to accurately detect paracetamol-induced liver injury and differentiate between DILI, healthy volunteers, and patients with non-hepatocellular organ damage [60]. However, when looked at individually, these three markers did not show superior diagnostics accuracy when compared to ALT [60]. These findings are questioning the results mentioned above that demonstrate high diagnostic accuracy also for the individual measurement of miR-122, GLDH and K18 [50,53].

In addition to the conventional mechanistic studies, proteomics has also been focused on. For instance, with the application of proteomics in the aforementioned MH cells, the cell adhesion molecule integrin subunit beta 3 (ITGB3) was identified as a possible marker for diclofenac-induced liver injury. ITBG3 was found to be expressed to a higher degree in MH cells generated from the blood of patients with diclofenac-induced DILI when compared to the control subjects (healthy subjects, patients with DILI by other drugs and patients with liver injury caused by alternative reasons) [61]. Moreover, mass-spectrometry-based quantitative proteomic assays have also been used in the search for DILI biomarkers, with which apolipoprotein E could be identified as a potential marker with high discriminatory power between DILI patients and healthy subjects [62].

Metabolomics is another promising approach toward the identification of DILI biomarkers. Different studies have revealed that mostly bile acids and glycerophospholipids, including glycocholic and tauroursodeoxycholic acid, show the most relevant metabolomic changes seen in DILI patients [63,64]. Interestingly, the metabolomic signature was also shown to correlate with the clinical DILI phenotype and the recovery status [64]. Metabolomics might therefore be useful not only for a better characterization of the type of DILI but also for the prediction of DILI outcome. Furthermore, a number of bile acids were found to correlate with more severe DILI, thus indicating that these bile acids might function as biomarkers both for diagnosis and severity prediction in DILI [63]. Regarding the use of metabolomics for differential diagnosis of liver injury, one study compared the metabolomic profiles of 248 samples from patients with nine different liver diseases and healthy controls and revealed that c-glutamyl dipeptides could serve as potential discriminatory markers [65]. The combination of ALT and γ-glu-citrulline in particular could distinguish DILI from other types of liver injury with an AUROC of 0.817 and 0.849 in the training and validation cohorts, respectively [65]. More recently, the metabolomic fingerprint of Polygonum multiflorum-induced liver injury was characterized and then compared to metabolomic pattern in patients with AIH and viral hepatitis [66]. In this study, a model using P-cresol sulfate vs. phenylalanine and inosine vs. bilirubin was highly accurate in differentiating PM-DILI from viral hepatitis and AIH [66].

In addition to analyzing potentially mechanistic biomarkers, novel diagnostic methods have been evaluated in the search for DILI biomarkers, such as genome-wide association studies (GWAS). GWAS have substantially increased the current knowledge regarding mechanistic biomarkers of DILI. As such, GWAS have identified HLA alleles to be associated with certain types of DILI, underlining the importance of the adaptive immune response in the pathogenesis of DILI [37,67,68,69]. Amoxicillin–clavulanate-induced DILI, for instance, has been associated with human leucocyte antigen (HLA) class II DRB1*15:01-DQB1*06:02 and HLA class I HLA-A*0201 [70,71,72]. Importantly, HLA-A*0201 highly correlated with DQB1*06:02 in HLA high-resolution genotyping of DILI cases and controls [70], indicating the importance of both HLA class I and II genotypes in the development of amoxicillin–clavulanate DILI. HLA-B*35-02, on the other hand, has been associated with minocycline DILI [73], and many more associations have been described [3]. However, while these HLA risk alleles have a high negative predictive value, the positive predictive value is low. Thus, HLA determination might help to exclude DILI caused by a certain type of medication. Yet, based on the current data, the use in diagnosis and more so as preventive genetic screening before initiation of drug treatment cannot be recommended [74].

## 6. Consortia Efforts for Biomarker Development in Drug-Induced Organ Injuries

In the search for more reliable safety biomarkers in clinical trials and diagnosis of medication-induced injury to the liver and other key organs (kidney, pancreas, vascular and central nervous system), different consortia have been formed, e.g., the Critical Path Institute’s Predictive Safety Testing Consortium (PSTC) in the United States and the Safer and Faster Evidence based Translation (SAFE-T) consortium in Europe [53] as well as more recently the TransBioLine (Translational Safety Biomarker Pipeline) consortium [75]. With the goal to minimize medication-induced risk to patients’ health, the latter is aiming at identifying and developing biomarkers for drug-induced organ injuries with regulatory qualifications by the FDA (U.S. Food and Drug Administration), EMA (European Medicine Agency) and PDMA (Pharmaceuticals and Medical Devices Agency). To this end, the TransBioLine consortium has built clinical study networks around Europe and the USA collecting patient data and samples. The study is currently organized into five organ system work packages, including a work package for DILI, and six enabling work packages, e.g., regarding liquid biopsy for miRNA measurement, biobanking, assay development and data management [75].

## 7. Prognostic Biomarkers—Recent Advances in DILI Research

In addition to the identification of diagnostic biomarkers, efforts have also been put into the detection of prognostic biomarkers for DILI outcome (Table 2). For instance, an international collaborative study including the above-mentioned SAFE-T consortium showed a strong association between K18, osteopontin (OPN) and macrophage-colony-stimulating factor receptor (MCSFR) and liver-related death or transplant within six months of DILI onset [53]. They further demonstrated that the performance of the Model for end-stage liver disease (MELD) score could be increased when it was used in combination with K18 and MCSFR. When compared to the MELD score alone, an improved specificity of 89% for severity prediction could be achieved in particular for MELD scores between 20 and 29 without a decrease in sensitivity (93%) [53]. On the other hand, with regards to paracetamol-induced liver injury, it was shown that out of miR-122, HMBG1, K18 and caspase-cleaved K18, only HMBG1 was predictive of coagulopathy and therefore of a more severe outcome [52]. However, since with death or liver transplantation vs. coagulopathy different endpoints were chosen, the comparability of the results across the mentioned studies is limited.

Regarding miRNA as prognostic DILI markers, the before-mentioned study by Dear et al. failed to demonstrate a prognostic value for miR-122 in paracetamol-induced DILI [52], while other studies have demonstrated that miRNA could be of prognostic nature as well. For instance, miR-122-5p alone or in combination with miR-382-5p has been shown to have a higher sensitivity regarding the prediction of paracetamol-induced ALF when compared to ALT [76]. Moreover, miR-122, miR-4463 and pre-miR-4270 were shown to be inversely correlated with a fatal outcome within six months of DILI onset [77]. miR-122, in particular, when combined with albumin could identify DILI patients with a fatal outcome within six months of DILI onset with a sensitivity and specificity of 100% and 81%, respectively [77]. Differences in patient cohorts, in particular regarding the causative agent (paracetamol vs. non-paracetamol) and choice of endpoints, most likely explain the conflicting results found in those studies.

Interestingly, metabolomics has also been applied in DILI severity prediction [78]. Together with five cytokines that were expressed at a significantly lower level in severe-DILI patients, 31 metabolites with a differential expression between patients with severe and non-severe DILI caused by mainly herbal and dietary supplements and antibiotics among others were identified. Pathway analysis showed that mainly the primary bile acid biosynthesis and alpha-linolenic acid pathways were altered [78]. A model comprising both these metabolites and cytokines could differentiate between severe and non-severe DILI with an AUROC of 0.983 [78].

The prognostic power of cytokines was also studied in patients from the US DILI network (DILIN) cohort, demonstrating that decreased levels of IL-9, IL-17, platelet-derived growth factor bb (PDGF-bb) and regulated on activation, normal T expressed and secreted (RANTES) could predict an unfavorable outcome, defined as death within 6 months of liver injury onset, with an accuracy of 92%. The prognostic accuracy could be even improved by the combination of these four analytes with hypoalbuminemia (accuracy 96%) [79]. However, when Bonkovsky et al. tried to validate these findings in a larger cohort from the DILIN and the Acute Liver Failure Study cohort, the only prognostic marker panel that could be identified was the combination of hypoalbuminemia and decreased levels of RANTES. Yet, while this combination of low albumin with a cut-off of 2.8 g/dL and low RANTES levels predicted six-month mortality with a specificity of 91%, sensitivity was rather low with only 39% [80].

**Table 2 ijms-23-06049-t002:** Overview of diagnostic and prognostic biomarkers in drug-induced liver injury.

Marker	Literature Reference	Summary
Proposed biomarkers/biomarker panels for DILI detection and or differentiation from other liver diseases		
miR-122	Thulin P, et al. Liver Int. 2014 [50] Howell LS, et al. Expert Rev Mol Diagn. 2018 [49] Rupprechter SAE, et al. Br J Clin Pharmacol. 2021 [47]	miR-122 as a liver-enriched miRNA has been observed before ALT elevation in patients with paracetamol overdose and has repeatedly been shown to be an early biomarker for DILI with higher sensitivity and specificity compared to ALT.
	Church RJ, et al. J Dig Dis. 2019 [53]	Controversially, a lower correlation of miR-122 with ALT compared to GLDH as well as a high inter- and intra-individual variability have been observed.
miR-129	Wang K, et al. Proc Natl Acad Sci USA. 2009 [54]	Another liver-enriched miRNA, which has been proposed as an early biomarker for DILI showing dose-dependent changes parallel to ALT in patients with paracetamol overdose.
	Liu XL, et al. Hepatology. 2020 [55]	Specificity for DILI is questioned by the observation that miR-129 could serve as a marker for NAFLD progression.
GLDH	Church RJ, et al. J Dig Dis. 2019 [53] Schomaker S, et al. PLoS One. 2020 [57] Roth SE, et al. Clin Pharmacol Ther. 2020 [58] Church RJ, et al. PLoS One. 2020 [81]	GLDH was shown to correlate better with ALT in DILI patients when compared to miR-122 and thus has been proposed as a promising biomarker for DILI detection. More specific marker for liver injury than ALT as demonstrated by a study in patients with Duchene muscular dystrophy. Only mildly influenced by age and gender with low intra- and inter-subject variability. Potential marker for mitochondrial damage or mitophagy.
	Harrill AH, et al. Clin Pharmacol Ther. 2012 [59] Roth SE, et al. Clin Pharmacol Ther. 2020 [58]	GLDH elevation might occur as consequence of biliary obstruction, congestive hepatitis or clinically non-significant liver injury following exposure to heparins, limiting the specificity for DILI.
K18	Thulin P, et al. Liver Int. 2014 [50]	The K18 marker M65 has been shown to not only increase earlier and to a higher extent in patients with paracetamol overdose than ALT but also to decline earlier after drug withdrawal.
K18, GLDH, miR-122	Llewellyn HP, et al. Toxicol Sci. 2021 [60]	The combination of full-length K18, miR-122 and HMBG1 predicted paracetamol-induced acute liver injury better than ALT. However, when evaluated individually, none of the markers showed better diagnostic accuracy in comparison to ALT.
miR-122, HMGB1, K18	Antoine DJ, et al. Hepatology. 2013 [51]	The combination of full-length K18, miR-122 and HMGB1 could predict ALI after paracetamol overdose before increase in ALT.
ITGB3	Dragoi D, et al. Front Pharmacol. 2018 [61]	ITGB3 was shown to be upregulated in MH cells from patients with diclofenac-induced DILI and is therefore proposed as a specific marker for this type of DILI.
Apolipoprotein E	Bell LN, et al. Aliment Pharmacol Ther. 2012 [62]	Potential diagnostic DILI marker with high diagnostic accuracy (AUROC 0.97).
γ-Glu-Citrulline	Soga T, et al. J Hepatol. 2011 [65]	Good differentiation of DILI and other types of liver injury in combination with ALT (AUROC 0.817).
Serum metabolites (bile acids)	Ma Z, et al. Medicine (Baltimore). 2019 [63]	Serum metabolites for bile acid synthesis, i.e., palmitic acid, taurochenodeoxycholic acid, glycocholic acid and tauroursodeoxycholic acid, were identified as possible diagnostic markers for DILI with a higher expression when compared to healthy controls.
Metabolomic classification model (P-cresol sulfate vs. phenylalanine and inosine vs. bilirubin)	Huang Y, et al. Front Med (Lausanne). 2020 [66]	A classification model consisting of P-cresol sulfate vs. phenylalanine and inosine vs. bilirubin could distinguish between PM-DILI, AIH and HBV with a diagnostic accuracy of 89.8% (sensitivity 92.3%, specificity 88.9%).
GWAS for genetic susceptibility	Lucena MI, et al. Gastroenterology. 2011 [70] Urban TJ, et al. J Hepatol. 2017 [73] Donaldson PT, et al. J Hepatol. 2010 [71] Andrade RJ, et al. Hepatology. 2004 [72]	Various GWAS have shown genetic susceptibility for specific types of DILI, e.g., HLA DRB1*15:01-DQB1*06:02 and HLA-A*0201 in amoxicillin–clavulanate DILI or HLA-B*35-02 in minocycline-induced liver injury. However, the positive predictive value of HLA risk alleles is comparably low impeding the use for DILI diagnosis.
Proposed biomarkers/biomarker panels for severity prediction in DILI patients in the clinical setting		
K18, OPN, MCSFR	Church RJ, et al. J Dig Dis. 2019 [53]	A strong correlation between K18, OPN and MCSFR and liver-related death or transplant within six months of DILI onset has been described.
HMBG1	Dear JW, et al. Lancet Gastroenterol Hepatol. 2018 [52]	In paracetamol-induced liver injury, out of miR-122, HMGB1 and K18, only HMGB1 was predictive of coagulopathy with a sensitivity of 88% at a specificity of 95%.
miR-122-5p (+/− miR-382-5p)	Vliegenthart AD, et al. Sci Rep. 2015 [76]	miR-122-5p alone or in combination with miR-382-5p has been shown to predict paracetamol-induced liver failure with higher sensitivity than ALT.
miR-122, miR-4463 and pre-miR-4270	Russo MW, et al. Liver Int. 2017 [77]	miR-122, miR-4463 and pre-miR-4270 inversely correlated with a fatal outcome within the first six months after DILI onset. miR-122 in combination with albumin could predict a fatal outcome with a sensitivity and specificity of 100% and 81%, respectively.
Serum metabolites (bile acids)	Ma Z, et al. Medicine (Baltimore). 2019 [63]	The serum metabolites glycocholic acid, taurocholic acid, tauroursodeoxycholic acid, glycochenodeoxycholic acid, glycochenodeoxycholic sulfate and taurodeoxycholic acid were shown to correlate well with more severe DILI.
IL-9, IL-17, PDGF-bb, RANTES	Steuerwald NM, et al. PLoS One. 2013 [79]	Lower levels of IL-9, IL-17, PDGF-bb and RANTES predicted a fatal outcome within six months of DILI onset with an accuracy of 92%. The accuracy of those four markers was even higher in combination with albumin (96%).
	Bonkovsky HL, et al. PLoS One. 2018 [80]	The predictive accuracy of IL-9, IL-17, PDGF-bb and RANTES could not be validated in a larger cohort from the DILIN and Acute Liver Failure Study cohort, in this cohort the only predictive panel was RANTES and albumin, which at lower levels predicted mortality within six months with a specificity of 91% at a low sensitivity of 39%.

Abbreviations: AIH: Autoimmune hepatitis, ALI: Acute liver injury; ALT: Alanine aminotransferase; AUROC: Area under the receiver operating characteristic; DILI: Drug-induced liver injury; DILIN: DILI network; GLDH: Glutamate dehydrogenase; GWAS: Genome-wide association studies; HBV: Hepatitis B virus; HLA: Human leucocyte antigen; HMBG1: High-mobility group box 1; IL: Interleukin; ITGB3: Integrin subunit beta 3; K18: Keratin 18; MCSFR: Macrophage-colony-stimulating factor receptor; miR: Micro-RNA; MH: Monocyte-derived hepatocyte-like cells; NAFLD: Non-alcoholic fatty liver disease; OPN: Osteopontin; PDGF: Platelet-derived growth factor; PM: Polygonum multiflorum Thunb; RANTES: Regulated on Activation, Normal T Expressed and Secreted.

## 8. Prediction of DILI—Where Do We Stand?

Not only diagnostic biomarkers, i.e., laboratory markers or panels used in a specific patient when DILI is suspected, but also markers for DILI prediction especially in drug development are highly sought after. In this regard, it is important to differentiate between patient and drug risk factors.

Regarding patient risk factors, for instance, it was recently demonstrated that pre-treatment levels of the inflammatory mediators interferon-gamma-induced protein 10 (IP-10) and soluble CD163 (sCD163) were associated with increased risk for DILI due to anti-tuberculosis agents and that IL-22 binding protein (IL-22BP) correlated with protection from DILI [82]. These mediators likely reflect the immune-inflammatory state of the patients before treatment initiation. IP-10 is released by different cell types, e.g., monocytes, fibroblasts, or endothelial cells, as a result of IFN-γ production and has been shown to be up-regulated in chronic inflammatory diseases [83,84,85]. IP-10 has already been associated with more severe DILI [78], highlighting the role of IP-10 in T-cell activation and trafficking [86]. sCD163 is seen as a biomarker for macrophage activation and inflammation [87] and has also been associated with chronic inflammatory states such as diabetes or chronic hepatitis-induced fibrosis [88,89]. These considerations indicate that a more activated inflammatory state prior to drug exposure increases the risk for DILI.

IL-22BP, on the other hand, is thought to inhibit the physiological effects of IL-22 in order to protect the body from exaggerated inflammation caused by IL-22 [90], which could explain the protective role in DILI. Conversely, IL-22BP has also been shown to prevent the adverse effects caused by IL-22 in cirrhotic patients and is therefore also thought to be protective of the development of acute or chronic liver failure [91].

While the above-mentioned patient-inherent risk factors are more likely to be used in the clinical setting, the identification of specific drug risk factors is particularly important during drug development. The detection of drugs with high DILI potential early on, preferably during the pre-clinical stages of drug development, could help to prevent the occurrence of DILI in the first place. In this regard, the DILI-sim Initiative, a public–private partnership comprising experts from academia, industry, and the FDA, has been implemented. This initiative has established a model called DILIsym for the prediction of liver adverse events of novel drug candidates during drug development by applying quantitative system toxicology [92]. The initiative also aims at improving the interpretation of liver biomarkers during clinical studies and with this potentially enhancing DILI diagnosis and management of adverse liver outcomes in clinical practice [92]. DILIsym focuses on the main pathomechanisms suspected behind DILI development: production of reactive metabolites, formation of reactive oxygen species (ROS), mitochondrial toxicity, dysfunction of bile export salt pumps (BSEP) resulting in accumulation of toxic bile acids, and activation of the innate immune system [92,93].

More recently, DILI-CAT, a novel data-driven algorithm for the identification of DILI early during drug development, has been proposed. This algorithm is based on DILI phenotypes and signatures caused by specific drugs, as characterized by latency, R-values, and AST/ALT ratio. It was shown that the DILI-CAT Score could discriminate DILI caused by ximelagatran and warfarin and could therefore also be of use in drug development schemes [94].

Regarding the evaluation of different toxicity mechanisms, the measurement of BSEP inhibition is an integral part of DILI prediction models in the pre-clinical setting since inhibition of these BSEPs plays an important role in DILI, in particular with a cholestatic pattern [95,96]. In vitro inhibition of BSEP by certain drugs, such as cyclosporin A, bosentan or troglitazone, has been demonstrated to correlate with the risk of these drugs for cholestatic DILI [97,98,99]. More recently, it was shown that by a fluorescein export assay, which analyzes both the inhibition of BSEP and of multidrug-resistance-associated protein 2 (MRP2), it was possible to distinguish between hepatotoxic and non-hepatotoxic agents [100]. Based on these findings, industry and medical regulatory agencies, such as the EMA and the FDA, have proposed to implement BSEP inhibition testing into drug development schemes [101,102,103]. However, the importance of testing BSEP inhibition was challenged by Chan et al. who, by using a compilation of DILI datasets, showed that the potential for BSEP inhibition was not associated with DILI potential and with this observation concluded that measurement of BESP inhibition in vitro cannot adequately predict DILI [104]. Noteworthy, data that have demonstrated a correlation between BSEP inhibition and DILI was mostly based on in vitro studies using high drug concentrations. Thus, it is possible that BSEP inhibition leads to the release of DAMPs and by this serves as an additional signal for the activation of an adaptive immune response rather than having a direct hepatotoxic effect [37,105].

Mitochondrial damage is also thought to play a pivotal role in DILI, with impairment of the mitochondrial respiratory chain and ß-oxidation of fatty acids, increased mitochondrial permeability and depletion of mitochondrial DNA being the main mechanisms involved in the pathogenesis of DILI [37]. These mitochondrial injuries lead to ATP depletion, overaccumulation of fatty acids and increased ROS formation resulting in oxidative stress [106]. Increased ROS production can also be a result of the depletion of antioxidants or generation of reactive metabolites by drug-metabolizing enzymes such as CYP450 [107,108]. The thus generated oxidative stress can cause further cell damage and consecutively contribute to liver damage. In line with these considerations, it was shown that drugs with the highest DILI concern, i.e., drugs with boxed warning or withdrawal from the market, were associated with dual inhibition of mitochondrial function and BSEP [109]. In vitro testing for mitochondrial damage has therefore been proposed as another predictive method for the evaluation of DILI potential during drug development [109]. However, contradicting these findings is the observation that typical drugs that are known to cause mitochondrial damage, such as metformin, linezolid or biguanides, only rarely cause relevant liver injury. Moreover, metformin as a typical mitochondrial toxicant does not increase the susceptibility toward the development of DILI by concomitant drugs [37]. Thus, mitochondrial damage per se might actually not play a major role in DILI evolution. However, same as BSEP inhibition, mitochondrial injury could lead to the production of DAMPs and ROS and therefore promote an adaptive immune response and by this DILI [110,111].

## 9. Predictive Human Models—The Future of DILI Detection?

Taking all these considerations into account, it becomes clear that in the search for predictive DILI models, many methods have been tested, but none of those methods have been proven to accurately anticipate DILI. Since drugs can cause immune activation via a large range of mechanisms, e.g., formation of neoantigens, release of DAMPS due to BSEP inhibition and/or mitochondrial damage or non-covalent interactions with the major histocompatibility complex (MHC)/T-cell receptor complex, testing for all the potential mechanisms simultaneously is near to impossible. Moreover, DILI can also be caused by the formation of anti-drug antibodies [112,113]. These considerations show the need for reliable predictive human models to study the complicated DILI mechanism with the goal to identify DILI prior to its appearance in clinical trials and/or in the post-marketing phase.

As a base for such predictive DILI models, traditionally primary human hepatocytes (PHHs) have been used. PHHs represent the in vivo metabolism, express phase I and II enzymes, retain CYP activity and can therefore reflect toxicity [23,114]. Human hepatic cell lines, such as the human-liver-derived HepaRG cell line, on the other hand, have the benefit of higher functional stability while retaining various liver-specific functions, including major CYPs as well as having an unlimited availability and propagation potential. However, PHHs show high phenotypic variability and instability regarding phenotypic characteristics in in vitro cultures, while liver cell lines are derived from liver cancer cells and therefore lack physiological drug-metabolizing enzymes and transporters limiting the representativeness of normal cellular toxic responses [23,114,115]. Moreover, PHH and hepatocyte cell lines constitute only simplified models lacking in vivo physiological processes, such as bile flow or continuous perfusion.

Alternatively, human pluripotent stem cells can be used, with both human embryonic stem cells (hESC) and induced pluripotent stem cells (iPSCs) having the ability to serve as progenitors for hepatocyte-like cells (HLCs). However, due to ethical restraints and the therefore limited availability of hESC, human iPSCs derived from reprogrammed somatic cells are mainly utilized [116]. Since iPSCs are pluripotent, they can serve as various cell lines in multicellular systems enabling the development of more physiological cell models [117,118]. Moreover, these iPSC-generated hepatocyte-like cells can retain hepatocytic function, such as serum protein secretion or drug metabolism. However, metabolic activity is slow when compared to physiological situations limiting the applicability of iPSCs in DILI models [119]. Efforts to enhance function of the iPSC-induced HLCs have been made, e.g., the generation of cellular polarity by using collagen matrices, the creation of a microenvironment with fluid flow, the exposition toward bile acid synthesis components or the supplementation with amino acids or a certain microbiome compound [120,121,122,123]. Organoids comprising iPSC-derived HLCs can then serve as three-dimensional cellular models for DILI prediction or in vitro modeling of disease mechanisms [124,125]. However, one caveat of iPSC-based liver models is the lack of biliary and vascular structures and therefore of a representative blood flow which leads to a lack of oxygen and chemical gradients. To overcome this limitation, efforts have been made to develop three-dimensional culture systems with vascular- and bile-canaliculi-like structures [126]. Interestingly, a human liver organoid model from human PSCs was recently developed that contained cells with hepatocytic function, such as the secretion of complement factors and albumin and also showed a bile-duct-comparable microarchitecture with reproducible bile acid transportation [127]. Using these organoids, DILI could be predicted with high accuracy: via a multiplexed analysis with 238 marketed drugs, viability, mitochondrial toxicity and/or cholestasis could be detected with high sensitivity and specificity of 89%, respectively [127]. Moreover, more sophisticated bioengineering projects, such as the “liver-on-chip”, are being developed, which in the long run might be able to predict drug toxicity long before DILI would appear as a relevant problem in clinical studies or post-marketing safety evaluations [128]. These bioengineered liver models provide near to physiological interaction of hepatocytes and stroma cells which can enhance hepatocyte functionality. However, these systems are complex, even more so if multi-organ models are developed, impeding compatibility with high-throughput screening and resulting in major challenges due to cellular interactions [114]. Additionally, predictive human models focus more on dose- or function-dependent DILI rather than on the individual susceptibility toward DILI limiting their application in the clinical setting and DILI diagnosis.

## 10. Conclusions

DILI remains a challenging diagnosis, and mistakenly suspected DILI can have major impacts on the individual patients as well as on drug development regimens. Improving DILI diagnosis with the identification of specific biomarkers has gained particular attention in recent years. In this review, different laboratory methods for DILI diagnosis are presented, mainly currently available in vitro causality assessment tools and serum biomarkers for diagnosis and severity prediction. Furthermore, laboratory tests for the prediction of patient inherent risk factors for certain types of DILI and predictive methods regarding drug inherent risk for DILI were discussed. This review shows that while major improvements have already been made, further research is necessary to overcome the challenges in DILI diagnosis and prediction.

## Figures and Tables

**Figure 1 ijms-23-06049-f001:**
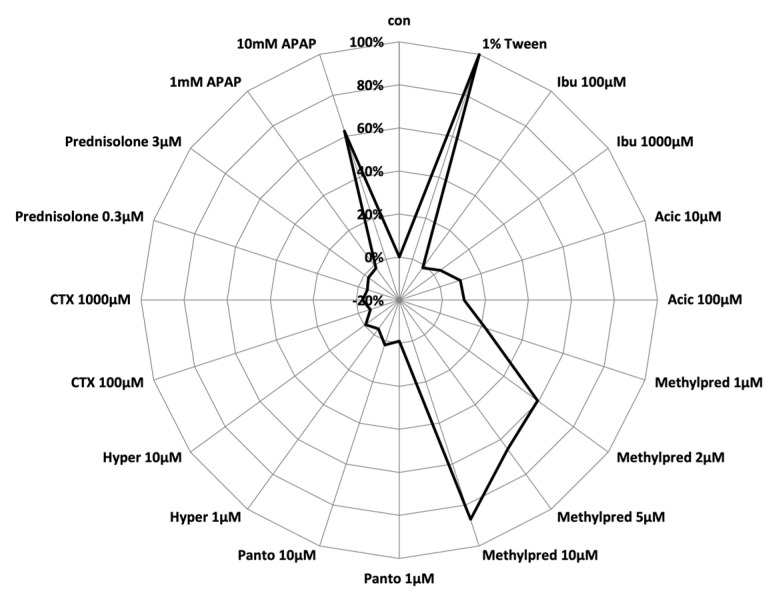
Example of an MH cell test result. The spiderweb graph shows the toxicity results in a patient with suspected DILI. Methylprednisolone as the likely cause of DILI in this case (causality assessment: highly likely with a positive rechallenge) induces toxicity of 60% at 2 × C_max_, of 66% at 5 × C_max_ and 87% at 10 × C_max_. No toxicity of the comedication with ibuprofen, acyclovir, pantoprazole, hyperforin (a herbal remedy), ceftriaxone or prednisolone, which was used to treat the DILI episode, was observed in MH cell testing in this patient. APAP is a standard part of the test for demonstrating dose-dependent drug-induced liver injury. Lysis with 1% TWEEN^®^20 (polyethylene glycol sorbitan monolaurate) is used as a positive control. Abbreviations: Acic: Acyclovir; APAP: Acetaminophen; con: Negative control; CTX: Ceftriaxone; Ibu: Ibuprofen; Hyper: Hyperforin; Ibu: Ibuprofen; Methylpred: Methylprednisolone; Panto: Pantoprazole.

**Table 1 ijms-23-06049-t001:** RUCAM—drug causality assessment.

Item 1: Time to Onset					Assessment ^2^
	Hepatocellular Type ^1^		Cholestatic Type ^1^		
	Initial treatment	Subsequent treatment	Initial treatment	Subsequent treatment	
○ From beginning of the drug: Suggestive Compatible	5–90 days <5 or >90 days	1–15 days >15 days	5–90 days <5 or >90 days	1–90 days >90 days	+2 +1
○ From cessation of the drug Compatible	≤15 days	≤15 days	≤30 days	≤30 days	+1
Note: If reaction begins before starting the medication or >15 days (hepatocellular)/>30 days (cholestatic) after stopping the medication, the injury should be considered unrelated and RUCAM cannot be calculated					
Item 2: Course	Change in ALT between peak value and ULN		Change in ALP (OR TBIL) between peak value and ULN		
After stopping the drug Highly suggestive Suggestive Compatible Inconclusive Against the role of the drug	Decrease ≥ 50% within 8 days Decrease ≥ 50% within 30 days Not applicable No information or decrease ≥ 50% after 30 days Decrease < 50% after 30 days or recurrent increase		Not applicable Decrease ≥ 50% within 180 days Decrease < 50% within 180 days Persistence or increase or no information Not applicable		+3 +2 +1 0 −2
If drug is continued Inconclusive	All situations		All situations		0
Item 3: Risk factors	Ethanol		Ethanol or Pregnancy (either)		
Alcohol or Pregnancy	Presence		Presence		+1
	Absence		Absence		0
Age	Age ≥ 55 years Age < 55 years		Age ≥ 55 years Age < 55 years		+1
					0
Item 4: Concomitant drug(s)					
None or no information or concomitant drug with incompatible time to onset Concomitant drug with suggestive or compatible time to onset Concomitant drug known to be hepatotoxic with a suggestive time to onset Concomitant drug with clear evidence for its role (positive rechallenge or clear link to injury and typical signature)					0 −1 −2 −3
Item 5: Exclusion of other causes of liver injury					
Group 1 (6 causes) Acute viral hepatitis due to HAV (IgM-HAV), or HBV (HbsAG and/or IgM anti-Hbc), or HCV (anti-HCV and/or HCV-RNA with appropriate clinical history) Biliary obstruction (by imaging) Alcoholism (history of excessive alcohol intake and AST/ALT-ratio ≥ 2) Recent history of hypotension, shock, or ischemia (within 2 weeks of onset) Group 2 (2 categories of causes) Complications of underlying diseases such as AIH, sepsis, chronic hepatitis B or C, primary biliary cholangitis or primary sclerosing cholangitis, or Clinical features or serologic and virologic tests indicating acute CMV, EBV or HSV			All causes in Group 1 and 2 ruled out		+2
			All causes in Group 1 ruled out		+1
			4 or 5 causes in Group 1 ruled out		0
			<4 causes in Group 1 ruled out		−2
			Non-drug cause highly probable		−3
Item 6: Previous information on hepatotoxicity of the drug					
Reaction labeled in the product characteristics Reaction published but unlabeled Reaction unknown					+2 +1 0
Item 7: Response to readministration:					
Positive	Doubling of ALT with drug alone		Doubling of ALP (or TBIL) with drug alone		+3
Compatible	Doubling of ALT with the suspect drug combined with another drug which had been given at the time of onset of the initial injury		Doubling of ALP (or TBIL) with the suspect drug combined with another drug which had been given at the time of onset of the initial injury		+1
Negative	Increase in ALT but less than ULN with drug alone		Increase in ALP (or TBIL) but less than ULN with drug alone		−2
Not done or not interpretable	Other situations		Other situations		0

^1^ Based on the R-value, which is defined as: (ALT/ULN)/(ALP/ULN), with R ≥ 5 defining a hepatocellular, R ≤ 2 a cholestatic and 2 < R < 5 a mixed-type injury. ^2^ Check one item only. Abbreviations: AIH: Autoimmune hepatitis; ALT: Alanine aminotransferase; ALP: Alkaline phosphatase; anti-Hbc: Anti-hepatitis B core antibody; AST: Aspartate aminotransferase; CMV: Cytomegaly virus; EBV: Epstein–Barr virus; IgG: Immunoglobulin G; IgM: Immunoglobulin M; HAV: Hepatitis A virus; HbsAg: Hepatitis B surface antigen; HBV: Hepatitis B virus; HCV: Hepatitis C virus; HSV: Herpes simplex virus; RNA: ribonucleic acid; RUCAM: Roussel Uclaf Causality Assessment Method; TBIL: Total bilirubin; ULN: Upper limit of normal.

## Data Availability

Not applicable.

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
