# Peer review of "Challenges and Future of Drug-Induced Liver Injury Research—Laboratory Tests"

_ijms, 2022, doi:10.3390/ijms23116049_

Round 1

Reviewer 1 Report

The topic of this review is interesting and appropriate.

IDILI is idiosyncratic in the sense that most patients will not develop IDILI when treated with a drug known to cause IDILI in some patients. Genetic studies have disclosed that a significant risk factor is an association with specific HLA genotypes. That provides very strong evidence that IDILI is immune mediated, but it is not the only mechanism. Metabolic idiosyncrasy also plays a role

Minor comments:

  • The subheading “DILI and autoimmune hepatitis: differences and overlaps in presentation and diagnosis”, should not better read “Drug induced autoimmune hepatitis and autoimmune hepatitis differences and overlaps in presentation and diagnosis”? I miss a brief description about the ontogeny of DI-AIH, maybe explaining how an initial DILI event my result in the loss of tolerance to self-antigens, and that this will continue even in the absence of the triggering drug
  • CD69 is a very useful biomarker for the lymphocyte transformation test (LTT), faster and more accurate that 3H Thymidine incorporation, yet no mention has been done.
  • Tc Lymphocyte activation is not the only mechanism in IDILI, and antidrug antibodies are also known (Kenna, J. G., Neuberger, J., and Williams, R. (1984). An enzyme-linked immunosorbent assay for detection of antibodies against halothane-altered hepatocyte antigens. J. Immunol. Methods 75, 3–14. doi: 10.1016/ 0022-1759(84)90219-9; Evidence of antibodies to erythromycin in serum of a patient following an episode of acute drug-induced hepatitis. M.J. Gómez-Lechón, J. Carrasquer, J. Berenguer and J.V. Castell. Clin. Exp. Allergy 26, 590-596 (1996). ISSN 0954-7894. https://doi.org/10.1111/j.1365-2222.1996.tb00581.x
  • Other non-immune related IDILI mechanisms are known, It has been proposed that mitochondrial injury is a common mechanism of IDILI; BSEP low expression is behind idiosyncratic cholestasis (Molecular mechanisms of hepatotoxic cholestasis by clavulanic acid in human hepatocytes. Petar D. Petrov, Polina Soluyanov, Sonia Sánchez-Campos, José V. Castell and Ramiro Jover. Food and Chemical Toxicology. Volume 158, December 2021, 112664 DOI: /10.1016/j.fct.2021.112664
  • Concerning the use of metabolomics and DILI, interesting progress has been made, (Metabolomic analysis to discriminate drug-induced liver injury (DILI) phenotypes. G. Quintás, T. Martínez-Sena, I. Conde, E. Pareja and J.V. Castell. Arch. Toxicol. (2021) DOI: 10.1007/s00204-021-03114-z.; Beger, R.D., J. Sun, and L.K. Schnackenberg, Metabolomics approaches for discovering biomarkers of drug-induced hepatotoxicity and nephrotoxicity. Toxicol Appl Pharmacol, 2010. 243(2): p. 154-66.

Reviewer 2 Report

In the manuscript presented, the Authors have described many methods to help diagnose DILI. However, the paper is slightly chaotic due to the multitude of topics covered. The following is a list of observations:

  1. line 33 - elevated ALP is not specific only to liver injury, which the authors do not mention
  2. the authors citing most of the studies do not indicate which drugs caused DILI (e.g. lines 131, 230, 304-310, 311-317) in each of these cases there is no information about the causative agent. Only in a few cases do they describe which drugs caused the liver damage. Also missing is a Summary Table, or an additional column in Table 2 to indicate the cause (specific drug) of the DILI and the indicators studied.
  3. line 237 - incorrectly written condition
  4. lines 183, 196, 212,302 - the pharmacopoeial name for acetaminophen is paracetamol. The authors use both names, but due to the greater popularity of the name paracetamol, this name should be used throughout the paper
  5. Line 302 - again, there is no specific information. What other drugs were used (non-paracetamol), as well as lines 302 and 303 - the information contained therein is too laconic and should be expanded
